# *Phytophthora plurivora*: A Serious Challenge for English Walnut (*Juglans regia*) Cultivation in Europe

**DOI:** 10.3390/microorganisms13092094

**Published:** 2025-09-08

**Authors:** Alessandra Benigno, Viola Papini, Federico La Spada, Domenico Rizzo, Santa Olga Cacciola, Salvatore Moricca

**Affiliations:** 1Department of Agricultural, Food, Environmental and Forestry Science and Technology (DAGRI), Plant Pathology and Entomology Section, University of Florence, 50144 Florence, Italy; viola.papini@unifi.it; 2Department of Agriculture, Food and Environment, University of Catania, 95123 Catania, Italy; federico.laspada@unict.it (F.L.S.); olga.cacciola@unict.it (S.O.C.); 3Laboratory of Phytopathological Diagnostics and Molecular Biology, Plant Protection Service of Tuscany, Via Ciliegiole 99, 51100 Pistoia, Italy; domenico.rizzo@regione.toscana.it

**Keywords:** disease management, nut and timber production, pathogenicity assays, phylogenetic analysis, phytopathogenic oomycete, polyphagous species, tree dieback

## Abstract

English walnut (*Juglans regia*) is a species that is highly valued for the quality of its wood and the nutritional and nutraceutical properties of its fruit. A severe dieback of *J. regia* trees was observed recently in orchards located in three geographically distinct areas of Tuscany, central Italy. Symptoms included root and collar rot, necrosis of the under-bark tissue, bleeding cankers, stunted growth, and crown dieback. Four *Phytophthora* species were obtained from 239 isolates found on symptomatic *J. regia* individuals. They were identified, on the basis of macro-morphological (colony shape and texture), micro-morphometric (shape and size of oogonia, antheridia, oospores, sporangia, and chlamydospores) and molecular (ITS sequencing) characters, as *P. gonapodyides*, *P. cactorum*, *P. citricola*, and *P. plurivora*. Among these species, *P. plurivora* was the species isolated with overwhelming frequency from symptomatic tissue and rhizosphere soil, suggesting it to be the putative etiological agent. Pathogenicity assays were conducted on 20 cm long detached *J. regia* branches for a definitive establishment of disease causation. Severe symptoms (extended necroses) were exhibited by branches infected with *P. plurivora*, proving its pathogenicity and high virulence on this host. The other *Phytophtora* species produced negligible necroses around the inoculation site. *P. plurivora* was recovered from all the investigated orchards, providing evidence that it is quite widespread. This study highlights the growing threat posed by the polyphagous *P. plurivora* to walnut cultivation and the sustainable business it fuels in Europe, underscoring the need for integrated management strategies to mitigate its economic and ecological impacts.

## 1. Introduction

English walnut (*Juglans regia* L.), a member of the genus *Juglans* (family Juglandaceae), is a tree species of significant global importance due to its ecological, economic, and nutritional/medicinal value. The range of this species, native to regions of Asia and Europe, extends from Central Asia, particularly the Himalayas, to the Middle East and south-eastern Europe [1]. In Europe, it is predominantly cultivated in the Mediterranean basin [2], where it plays a key role in local agro-ecosystems, contributing substantially to global walnut production. Here, in addition to English walnuts, other nut crops such as almond (*Prunus amygdalus*), chestnut (*Castanea sativa*), hazelnut (*Corylus avellana*), and pistachio (*Pistacia vera*) are commercial crops of great economic significance.

However, the cultivation of *J. regia* is currently in sharp decline, being increasingly threatened by biotic and abiotic stress factors, including emerging plant diseases. On the Italian peninsula, for example, its cultivation suffered a drastic plunge after the 1970s, shifting Italy from a leading producer country to a major importer [3]. Despite this negative trend, recent efforts have revitalised walnut production, particularly in central and northern Italy, through the establishment of specialised orchards, the adoption of advanced agronomic techniques, and mechanised harvesting [3,4].

Among the major threats to *J. regia* are fungal pathogens, such as *Ophiognomonia leptostyla*, the causative agent of walnut anthracnose, and, more recently, *Geosmithia morbida*, the agent responsible for Thousand Cankers Disease (TCD) in black walnut (*J. nigra*) [5,6]. This mitosporic ascomycete, vectored by the walnut twig beetle *Pityophthorus juglandis*, has in fact already proven its ability to host-jump on the native *J. regia* [7], favoured by the numerous mixed plantations of the two species that have been established in Europe with EU funds [8]. Additionally, a plethora of members of the Botryosphaeriaceae family have been widely implicated in canker and shoot blight diseases in walnut-growing regions worldwide. In California, for example, several Botryosphaeriaceae species, including *Botryosphaeria dothidea*, *Diplodia mutila*, *Diplodia seriata*, *Dothiorella iberica*, *Lasiodiplodia citricola*, *Neofusicoccum mediterraneum*, *Neofusicoccum nonquaesitum*, *Neofusicoccum parvum*, *Neofusicoccum vitifusiforme*, and *Neoscytalidium dimidiatum*, have been associated with a variety of symptoms like stem and shoot cankers, twig and fruit blight, necrotic leaf lesions, and graft union infections [4,9,10,11,12].

An additional, serious risk to walnut cultivation is posed by emerging oomycetes of the genus *Phytophthora*, destructive pathogens known for causing root rot and decline/dieback syndromes to a number of woody crops, as well as to natural and cultivated forests worldwide [13].

Symptoms indicative of a possible *Phytophthora* infection, such as general decline with reduced vigour, bleeding cankers, crown transparency, and die-back, were observed during 2023 and 2024 in some *J. regia* plantations in Tuscany, central Italy. Laboratory isolations revealed some *Phytophthora*-like vegetative mycelia to be frequently associated with underbark necrotic tissue. Baiting of soil/root samples taken at the base of symptomatic trees provided evidence that these matrices were heavily infected, with *Phytophthora* propagules (motile zoospores released from sporangia) attracted by living baits within 24 h. These findings prompted an in-depth investigation to elucidate the identity, pathogenicity, and role of the oomycete(s) possibly involved in the above syndrome. This study aimed to do the following: (1) identify the microbial agent(s) associated with the dieback of *J. regia* trees in plantations in Tuscany by means of macro- and micro-morphological, physiological, and molecular analyses (ITS sequencing); (2) assess the pathogenic potential and virulence of the isolates through controlled inoculation assays on host tissue; and (3) estimate the possible impact of the disease on the sustainability of *J. regia* cultivation in the future.

## 2. Materials and Methods

### 2.1. Survey Design and Sampling

Surveys were conducted during two consecutive growing seasons (from the beginning of spring 2023 to the end of summer 2024) in three 25-year-old *J. regia* populations originating from seed orchards and established for timber production, located in three geographically distinct areas of Tuscany (43.78624, 11.159425; 43.583187, 10.801350; 43.540960, 11.470407), to detect symptomatic tree individuals and monitor disease progression (Table 1). Tree canopies were visually inspected for symptoms like shoot blight and branch dieback; the trunks were monitored for bark necroses and emission of mucilage flow in their basal portion and at the root collar. Individuals exhibiting the above symptoms were further investigated for fine root necroses and root rot. The disease incidence (DI) and mortality rate (M) at each site were quantified along a 50-metre transect using the following methodology [14]:DI (%) = (n/N) × 100 and M (%) = (d/N) × 100
where n = the number of symptomatic trees, d = the number of dead trees, and N = the number of trees surveyed.

### 2.2. Phytophthora Isolation and Morphological Characterisation

A total of 210 samples were collected from 202 symptomatic *J. regia* trees, including bark portions and coarse and fine roots, yielding 239 isolates, including *P. plurivora* (175), *P. cactorum* (43), *P. citricola* (11), and *P. gonapodyides* (10). All samples were thoroughly washed, surface-sterilised in 1% NaClO for 2 min, immersed in 50% ethanol for 30 s, rinsed with sterile distilled water, air-dried, and subsequently placed on selective PARPN V8-agar medium. The medium was supplemented with 10 μg/L pimaricin, 200 μg/L ampicillin, 10 μg/L rifampicin, 25 μg/L pentachloronitrobenzene, and 50 μg/L nystatin. Plates were incubated in darkness at 24 °C for 24–48 h, and pure cultures were obtained by transferring individual hyphal tips onto V8-juice agar (V8A), malt extract agar (MEA), and potato dextrose agar (PDA). In addition, 30 soil samples were collected following the protocol of Benigno et al. [14].

Baiting was carried out at ambient laboratory temperature by submerging 200 mL of soil subsamples in 800 mL of distilled water in plastic containers and baiting them with fresh, healthy leaves of *Quercus suber* L., *Viburnum tinus* L. and *Rosa chinensis* Jacq. Necrotic lesions developed on the bait leaves within 2–3 days of incubation. Symptomatic leaf tissues were aseptically excised and placed onto V8-PARPN medium. Within 2–3 days, white, coenocytic, soft, and slow-growing mycelium grew on the selective medium. Isolates were then transferred onto V8A medium (100 mL of V8 juice, 2 g of CaCO_3_ and 20 mL of agar per L), as reported in Jung et al. [15]. Thereafter, isolates were grouped into four distinct morphotypes on the basis of colony phenotypes and micro-morphological characteristics. In order to ascertain the cardinal growth temperatures, isolates were cultivated on V8A plates at temperatures of 5, 10, 15, 20, 25, 30, and 35 °C (with a margin of error of ±0.5 °C), in darkness, with a total of seven replicates per isolate. The measurement of colony diameters was conducted on a daily basis over a period of seven days. The reproductive structures of 40 selected isolates representing the four morphotypes were examined after 20 days of incubation in darkness at 20 °C on carrot agar (CA). Morphological structures, including sporangia, chlamydospores, hyphal swellings, oogonia, and antheridia, were observed and measured using an optical microscope (DM750; Leica Microsystems, Wetzlar, Germany) equipped with LAS EZ software version 2.0. For each isolate, sporangial morphology (n = 50) and the ability to form hyphal swellings and chlamydospores were documented. Morphometric data were obtained by measuring 50 structures per sample at 400× magnification. Based on cultural and micro-morphometric traits, isolates were identified as *P. gonapodyides* (H. E. Petersen) Buisman [16], *P.* cactorum (Lebert & Cohn) J. Schröt. [17], *P. citricola* Sawada [18], and *P. plurivora* Jung and Burgess [19]. Representative morphotypes were preserved on CA medium in the culture collection of fungi and oomycetes held at the Plant Pathology section (DAGRI), University of Florence, Italy.

### 2.3. DNA-Based Typing and Phylogenetic Analysis

Forty selected isolates representing the four morphotypes were transferred onto sterile cellophane overlays, placed on 9 cm diam Petri dishes filled with potato dextrose agar (PDA), and then incubated in the dark at 24 ± 1 °C for seven days. The mycelium was collected from the surface of the cellophane, and the DNA was extracted, purified, and stored according to Benigno et al. [14]. The non-coding ITS (internal transcribed spacer) region of the ribosomal RNA operon, which includes the internal 5.8 S gene and the flanking ITS1 and ITS2 spacers, was PCR-amplified using the ITS1-F (forward primer) and ITS4-R (reverse primer) primers of White et al. [20]. The PCR was carried out in a reaction volume of 25 µL containing PCR Buffer (1×), dNTP mix (0.2 mM), MgCl_2_ (1.5 mM), forward and reverse primers (0.5 µM each), Taq DNA Polymerase (Sigma Aldrich, St. Louis, Missouri, USA) (1 U), and 100 ng of DNA template. PCR cycling conditions were denaturation at 94 °C for 3 min, followed by 35 cycles of 94 °C for 30 s, 55 °C for 30 s, and 72 °C for 30 s; then, a final extension was conducted at 72 °C for 10 min. The NucleoSpin^®^ Gel and PCR Clean-up kit (Macherey-Nagel, Düren, Germany) were used to purify the PCR products, according to the manufacturer’s instructions. The analysis of nucleotide sequences was conducted with the programme FinchTV version 1.4.0 (Geospiza, Inc., Seattle, WA, USA). The investigation identified the morphotypes as *Phytophthora gonapodyides*, *P. cactorum*, *P. citricola*, and *P. plurivora*, based on a comparative analysis of their consensus sequences with those already deposited in the GenBank database (National Center for Biotechnology Information—NCBI). This analysis, conducted by applying the Basic Local Alignment Search Tool (BLAST) (https://www.ncbi.nlm.nih.gov/BLAST/, accessed on 5 December 2024), confirmed the identifications previously made through traditional morphological methods. The obtained sequences were submitted to and deposited in the NCBI GenBank database (see Table 2 for accession details). Sequence alignment was conducted with ClustalX v. 1.83 [21], as reported in Bregant et al. [22]. MEGA-12 software was used to perform Maximum Likelihood (ML) analyses, with all gaps included in the analysis. The software automatically determined the optimal model of DNA sequence evolution.

### 2.4. Pathogenicity Assays

The virulence of *P. gonapodyides*, *P. cactorum*, *P. citricola*, and *P. plurivora* was assessed by inoculating freshly detached branches of *J. regia* taken from trees cultivated in a plantation in Florence, Tuscany (43.78624, 11.159425). A total of 100 healthy branches of approximately 2–3 cm in diameter were taken from the lower parts of the crown of 20 trees (5 branches per tree) and cut into segments (150 in total) of approximately 20 cm in length, by selecting the regions between the whorls. The cut ends of branch sections were immediately dipped in melted paraffin wax to prevent desiccation. In the pathogenicity assay, *P. gonapodyides*, *P. cactorum*, *P. citricola*, and *P. plurivora*, as well as sterile plugs (control), were inoculated onto the 20 cm detached segments, for a total of 30 replicates per *Phytophthora* species and 30 for the control. With a sterile scalpel, a ~0.7 cm square lesion was made on each detached branch, and the bark was carefully excised. A 5 mm^2^ mycelial disc, obtained from cultures actively growing on carrot agar (CA), was placed over each wound. The inoculation sites were wrapped in sterile, water-moistened cotton wool, covered with aluminium foil, and sealed with electrical tape. Control detached branches were inoculated with sterile CA discs using the same procedure. After inoculation, the branches were placed in sealed plastic bags and incubated at approximately 20 °C. The assay was conducted for two months, in March and April 2025. Following the incubation period, the bark surrounding the inoculation sites was carefully peeled off, and lesion lengths were measured. Colonies emerging from infected tissues were subcultured onto CA and incubated at 20 °C in darkness. Identification of the isolates was performed through morphological characterisation (macroscopic and microscopic analyses) and molecular analysis of the internal transcribed spacer (ITS) region via PCR amplification and sequencing.

## 3. Results

### 3.1. Symptomology and Disease Incidence

Symptoms suggestive of *Phytophthora* spp. infection were observed in 305 *J. regia* individuals sampled in the three investigated plantations in the two-year period of 2023–2024 (Figure 1). Affected trees, aged 20–30 years, showed bleeding cankers (dark fluid oozing from the bark) on the lower stem (Figure 1a), collar root lesions, and leaf chlorosis. Necrotic lesions developed primarily above the soil surface at the root collar junction (Figure 1b) and progressively extended 30–50 cm upwards on the lower trunk. Bleeding cankers consisted of abundant exudation of mucilage, initially light in colour, then darkening as the infection progressed (Figure 1c,d). Tongue-shaped, orange-brown lesions of the inner bark, along with dark brown discoloration of the cambium and adjacent xylem tissue, were also observed (Figure 1e,f). A census of symptomatic trees revealed a disease incidence of 90%, with site-specific mortality rates ranging from 15% to 45%. Lesions were confined to cortical and phloem tissues, with limited penetration into the sapwood.

### 3.2. Morphological Characteristics of Phytophthora Isolates

Direct isolation was performed from rotten roots, soil samples, and the basal bark of symptomatic *J. regia* trees randomly selected in the three plantations. *Phytophthora* species were consistently recovered across all sites (Table 1).

Isolates exhibited distinct colony morphologies, which were clearly noticeable on solid culture media. Colonies of *P. plurivora* cultivated on V8A medium exhibited a petaloid growth pattern with sparse aerial mycelium, whereas moderate aerial hyphal development was observed on PDA. In liquid culture, sporangia were abundant, predominantly terminal, with occasional lateral or intercalary formation. These sporangia were semi-papillate and displayed diverse morphologies, ranging from ovoid to obpyriform and lemon-shaped, with an average size of 47–51 × 29–36 μm and a length-to-breadth ratio of 2 ± 0.2. Sporangiophores had an irregular, loosely branched structure. The occurrence of homothallic reproduction was evidenced by the presence of globular or sub-globular oogonia with a mean diameter of 27 ± 2 μm. Furthermore, oospores were either plerotic or aplerotic, with a wall thickness of 2 ± 0.2 μm. Paragynous antheridia measured 12 ± 2 × 9 ± 2.5 μm. No chlamydospores or hyphal swellings were detected across the isolates. Optimal mycelial growth was observed at 25 °C, with no development occurring beyond this temperature. Colonies of *P. citricola* exhibited a chrysanthemum-like pattern on V8 agar, PDA, and MEA, with sparse aerial mycelium on PDA. Growth occurred between 5 °C and 30 °C, with an optimum at 24 °C. Sporangia and oospores were produced abundantly on V8 agar. Sporangia were semipapillate, persistent, and displayed various shapes, measuring 28–78 × 21–47 µm. Bipapillate sporangia were also present and developed on single or sympodially branched sporangiophores. Hyphal swellings and chlamydospores were absent. The sexual phase was homothallic, with smooth-walled oogonia that typically had a rounded base, though occasionally a tapered base was observed. Paragynous antheridia, rarely amphigynous, were spherical to club-shaped. Oospores were predominantly plerotic with thick walls. *P. cactorum* produced colonies without a distinct pattern on V8, PDA, and MEA. Growth occurred within a temperature range of 5 °C to 30 °C, with an optimum at 25 °C. Sporangia were papillate and caducous, with short pedicels (<4 µm), and varied in shape—ellipsoidal, obpyriform, ovoid, or globose—measuring 24–50 × 19–36 µm. They developed on sporangiophores that were either simple or arranged in close or lax sympodia. Hyphal swellings were absent. Chlamydospores were globose, occurring both terminally and intercalarily, and measured 19–51 µm in diameter. The species was homothallic, with smooth-walled, hyaline oogonia measuring 18–38 µm in diameter. Paragynous antheridia were spherical to club-shaped and typically attached near the oogonial stalk. Oospores were either plerotic or aplerotic. Colonies of *P. gonapodyides* developed a rosette-like growth pattern on PDA, V8, and MEA within 7 days. Mycelial growth occurred between 5 °C and 30 °C, with optimal development at 25 °C. Sporangia and hyphal swellings were exclusively formed in non-nutrient liquid cultures. Sporangia were nonpapillate, persistent, and exhibited ellipsoid, ovoid, or distorted shapes with a tapered base, measuring 31–69 × 21–37 µm. These structures showed both internal extended and nested proliferation and were produced on unbranched sporangiophores. Hyphal swellings were globose, subglobose, or elongate, occurring singly or in chains (catenulate). Chlamydospores were absent. No sexual structures were observed, indicating a sterile sexual phase.

DNA sequencing confirmed the identity of representative morphotypes as *P. gonapodyides*, *P. cactorum*, *P. citricola*, and *P. plurivora*. BLAST searches revealed 100% homology between the isolates and reference sequences of the four species deposited in the GenBank database (Table 2).

The ITS-generated sequences were edited and aligned with representative isolates from clade 2 for *P. plurivora* and *P. citricola*, from clade 1 for *P. cactorum*, and from clade 6 for *P. gonapodyides*. Isolates formed well-supported clades, clustering with sequences from ex-type cultures (Figure 2).

### 3.3. Pathogenicity Assay

At the end of the artificial inoculation trial, all 20 cm long detached branches inoculated with *P. plurivora* exhibited a dark brown discolouration of the inner bark above and below the inoculation point and along their entire length (Figure 3). The pathogen caused complete necrosis of the plant tissue, to the extent that thirty days after inoculation it was not possible to measure beyond the extent of necroses because the samples were completely necrotised. In contrast, the control and the branches inoculated with *P. gonapodyides*, *P. cactorum*, and *P. citricola* showed no signs of disease, apart from a slight light-brown discoloration around the inoculation site. The lesions observed on the branches inoculated with *P. plurivora* were confirmed to be caused by this oomycete, as it was successfully re-isolated from different points along the necrotic tissue. The other *Phytophthora* species were recovered only from the necrotic edge around the inoculation site. Identification of inoculated isolates was carried out by morphological characterisation (both macroscopic and microscopic) and molecular analysis of the internal transcribed spacer (ITS) region using PCR amplification and sequencing.

## 4. Discussion

Here, we provide evidence that *P. plurivora* is the agent responsible for inducing collar and basal trunk necroses, exudation of dark, tarry mucilage on the basal stem, canopy transparency, and general decline/dieback of *J. regia* trees in some Italian plantations. The lesion areas and the lesion expansion rates of *P. plurivora* over the detached branches of *J. regia* observed in the inoculation trials prove the high virulence of this oomycete on this host and leave no interpretative doubts about the aetiology of the dieback observed in the field.

The severe dieback induced by *P. plurivora* poses a serious challenge to *J. regia* cultivation in major walnut-growing areas of Europe, especially in Mediterranean countries, where its impacts could be significant for both agricultural productivity and ecosystem conservation. In Italy, *J. regia* is a keystone species within agricultural and ecological landscapes, supporting a high-value walnut (fruit and timber) industry and providing essential ecosystem services, such as biodiversity conservation, soil stabilisation, and carbon sequestration [23,24,25].

*P. plurivora* is commonly associated with tree diebacks in forest ecosystems. It has been isolated from a plethora of woody hosts (e.g., *Abies alba*, *Alnus* spp., *Betula pendula*, *Carpinus betulus*, *Castanea sativa*, *Fagus sylvatica*, *Pinus sylvestris*, *Quercus* spp., *Tilia* spp., etc.), both in natural forests and in tree plantations; it thrives especially in wet environments, where it also affects various undergrowth shrub species such as *Cornus mas*, *Hedera helix*, *Ilex aquifolium*, *Sambucus* spp., etc. [19,26,27]. The documented history of *P. plurivora* in causing widespread decline in European forests, particularly in *Quercus* spp., underscores its destructive potential. On oak hosts, the pathogen has been associated with fine root deterioration, increased drought sensitivity, and secondary pest infestations [28,29,30,31,32].

Since the late 1990s, the pathogen has also been increasingly linked to elevated levels of mortality in European beech forests. In these ecosystems, the pathogen contributes to crown transparency, reduced foliage size, and progressive canopy dieback [19,33,34,35].

The high aggressiveness of *P. plurivora* towards the major European tree species (e.g., *F. sylvatica*, *Q. robur*, *Alnus glutinosa*, and *Acer platanoides*) and its wide implication in forest decline suggests a lack of co-evolution with these native hosts. The most likely hypothesis is that *P. plurivora* was introduced to Europe through infected plant stocks and then became established through its persistent presence in nurseries [36].

However, *P. plurivora* is also increasingly reported as an emergent and significant pathogen in a range of non-forest hosts. It has been reported as causing damaging cankers and substantial fine-root losses in *Malus domestica* and *Malus sieversii* [19,26]; it has been shown to induce general decline in mature *Juglans nigra* orchards [37]; it has been isolated from rotten roots of ornamental *Taxus baccata* [19]; it has been associated with root rot and sudden mortality in *Olea europaea* [27]; and it has been reported as a serious threat to *Rhododendron* spp. production in nursery systems [38].

The large host range of *P. plurivora* is also likely underestimated, due to some taxonomic uncertainty about its identity, which has meant that in the past, attacks by this pathogen were, in some cases, attributed to other *Phytophthora* species. Indeed, prior to the taxonomic revision of Jung and Burgess [19], this oomycete had been reported either as *Phytophthora citricola* [39] or as *Phytophthora inflata* [40]. The inappropriate determination of the identity of this *Phytophthora* for some time may therefore have lessened a proper assessment its impact and role in agro-forestry systems.

The severity of the symptoms (gradual loss of vigour, with progressive death of twigs and branches, often culminating in the mortality of the tree) caused by *P. plurivora* on *J. regia* suggests that contemporary environmental conditions are particularly conducive to disease. *P. plurivora* is, in fact, reported to be more competitive than other congeneric species in semiarid–wet soil [37]. Climate change, with rising temperatures and changing precipitation patterns, particularly with the increase in extreme rainfall events that, even in the arid areas of southern Europe, occasionally cause extensive flooding, is reshaping plant–pathogen interactions. In the Mediterranean basin, an area recognised as a climate change hotspot, climate anomalies are creating favourable conditions for the increased proliferation and spread of oomycete pathogens like this one, increasing host susceptibility, with negative implications for agricultural and forestry systems [41]. When establishing new plantations in this area, it would be advisable to choose the site carefully, avoiding new plantations in low-lying areas, which are periodically subject to flooding and waterlogging.

Furthermore, anthropogenic factors, such as intensive agricultural practices, may enhance disease development. For example, the large-scale cultivation of genetically homogeneous crops at high densities [42], as is the case with some artificial plantations of *J. regia*, creates an environment that is conducive to both the emergence and spread of pathogens [43]. Moreover, factors such as soil compaction and waterlogging contribute to this process, providing favourable conditions for pathogens like *P. plurivora* to cause root and collar rot [44,45].

In Italy, *J. regia* has traditionally been grown throughout the peninsula. However, a marked decline in walnut cultivation has been observed, from 33,000 ha in the 1960s to 6,130 ha in 2023 [46]. Consequently, in order to meet a growing demand, Italy has been among the top five importing countries of in-shell walnuts since the mid-1970s [47]. Despite the considerable challenges confronting the walnut industry in Italy, it continues to represent a segment of the nation’s agricultural economy. In 2023, Italy produced 14,960 tonnes of in-shell walnuts [46], demonstrating its continued importance, particularly in terms of its contribution to rural livelihoods. This transition from a producer to a significant importer of walnuts represents a major change in Italy’s position. This phenomenon is indicative of broader trends in agricultural land utilisation and the challenges confronting the industry. The marked decline in cultivation area can be ascribed to a number of factors, including economic pressures, evolving agricultural practices, and the increasing prevalence of pests and diseases [48,49]. The increasing mortality of walnut trees resulting from *P. plurivora* may pose a further significant threat to the stability of the industry and cause a further reduction in the walnut production area. Beyond the economic implications, the potential disappearance of *J. regia* from both natural and cultivated landscapes could lead to serious ecological consequences. The loss of this species could lead to the degradation of local habitats, alterations to soil structure, and disruption to important ecological processes [50,51].

In order to mitigate the impact of *P. plurivora*, a multidisciplinary approach is required. Early detection is essential to prevent large-scale outbreaks, and molecular diagnostic tools such as PCR and its derivatives [e.g., nested PCR, real-time PCR, RFLP, AFLP, next generation sequencing, loop-mediated isothermal amplification (LAMP), and recombinase polymerase amplification (RPA) technology] can facilitate accurate and rapid identification of the pathogen [2,6,52,53]. The integration of molecular diagnostics into regular monitoring programmes would facilitate timely intervention and containment strategies [54].

In addition, the implementation of cultural practices, such as optimised drainage, reduced soil compaction, and improved sanitation measures, should be prioritised within an integrated disease management framework [45,55].

In the long term, the development of genetically resistant *J. regia* cultivars should be a key research priority, as has been attempted for *P. cinnamomi* [56]. The success of breeding programmes to enhance resistance to *P. plurivora* will require collaboration between researchers, breeders, and industry stakeholders.

Furthermore, there is a necessity for additional investigation into the environmental and biological factors that influence the virulence of *P. plurivora*, in order to support the development of precise management strategies. Such strategies might include the optimisation of irrigation techniques and the utilisation of environmentally sustainable biocontrol agents [57,58].

Finally, it is imperative to emphasise the critical role of policy support and stakeholder engagement in addressing this emergent threat. Increased investment in fundamental research, the development and refinement of diagnostic tools, and the implementation of comprehensive educational programmes are essential to ensure the adoption of optimal disease prevention and management strategies.

## 5. Conclusions

Previous studies have demonstrated that *P. plurivora* is involved in the deterioration of several important European forest species, but also its capacity to thrive outside forested environments. The extensive impact of *P. plurivora* on diverse crop plant species also highlights its significance as a pressing agricultural challenge. This is confirmed by the virulence of *P. plurivora* and its pivotal role in the dieback of *J. regia* orchards. This oomycete’s ability to infect multiple tree species, combined with its high environmental adaptability, suggests that its occurrence in *J. regia* orchards may represent only one aspect of a broader ecological issue. This scenario highlights the need for integrated monitoring programmes encompassing both agricultural and forest ecosystems to assess the overall impact of *P. plurivora* and develop effective long-term management strategies. Management efforts to reduce the spread of this emerging, polyphagous pathogen are urgently needed, especially in climate-vulnerable regions where nut production is reviving, such as the Mediterranean basin.

## Figures and Tables

**Figure 1 microorganisms-13-02094-f001:**
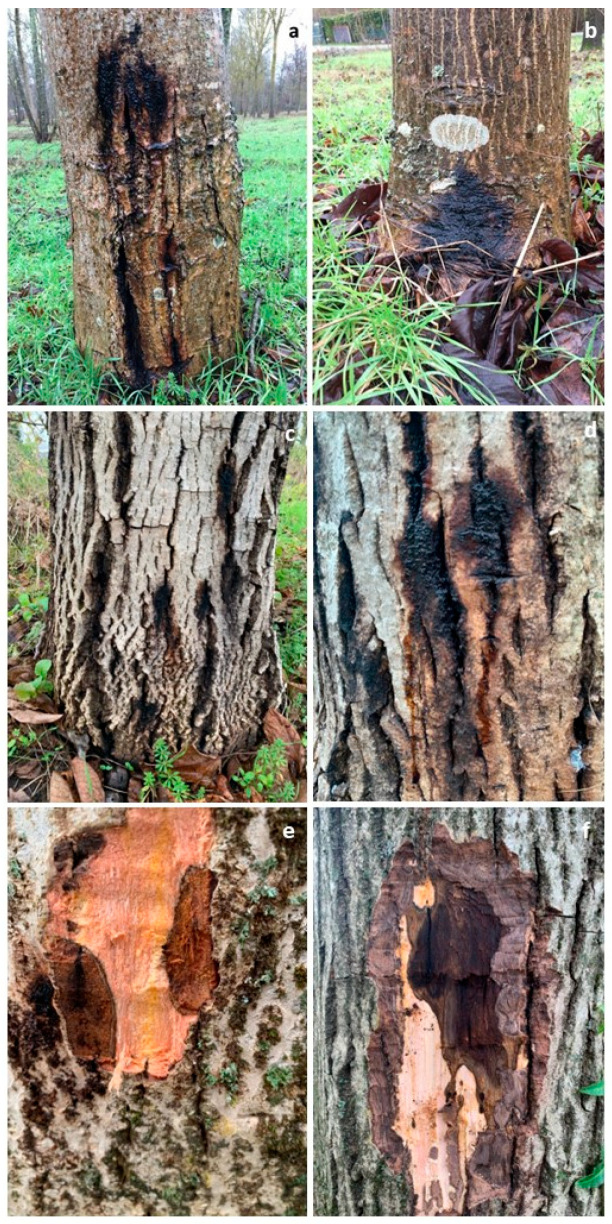
Symptoms of *Phytophthora* infection on the basal stem portion of 20–30-year-old *Juglans regia* trees in some orchards. Dead bark areas with dark mucilage flow (**a**); collar root lesion with dark mucilage flow (**b**); bleeding cankers on the lower trunk, with dark mucilage secretions from multiple bark cracks (**c**); close-up view of lesions actively exudating a dark tarry mucilage (**d**); particular of tongue-shaped brown necroses on the inner bark of the lower stem of *J. regia* trees (**e**,**f**).

**Figure 2 microorganisms-13-02094-f002:**
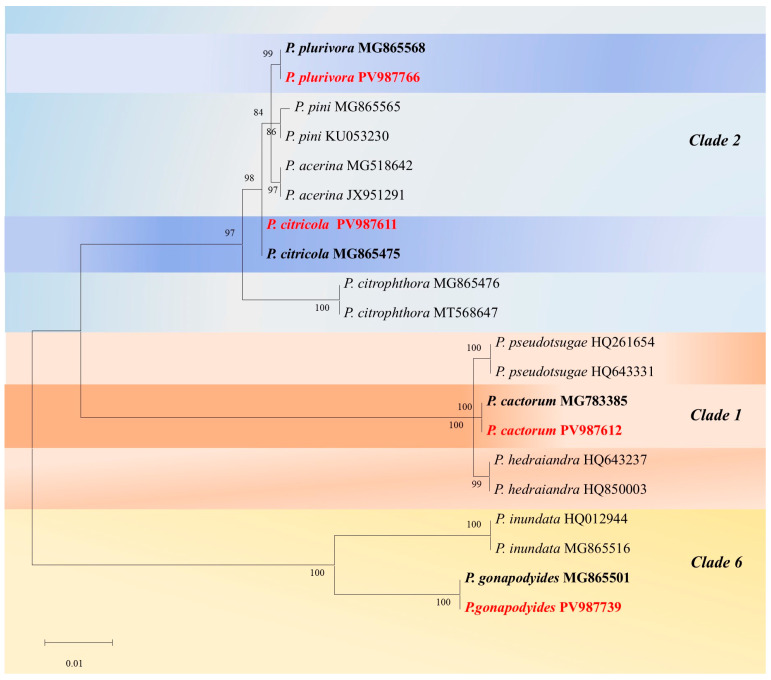
A phylogram generated by Maximum Likelihood analysis of the aligned set of ITS sequences of *Phytophthora* species belonging to three clades. The analysis used the General Time Reversible (GTR) model with a discrete Gamma distribution to model among-site rate heterogeneity. In the tree, depicted to scale, branch lengths represent the number of substitutions per site. Bootstrap values in percentages (1000 replicates) are shown on the nodes. Ex-type cultures and isolates resulting from this study are shown in bold and in red, respectively.

**Figure 3 microorganisms-13-02094-f003:**
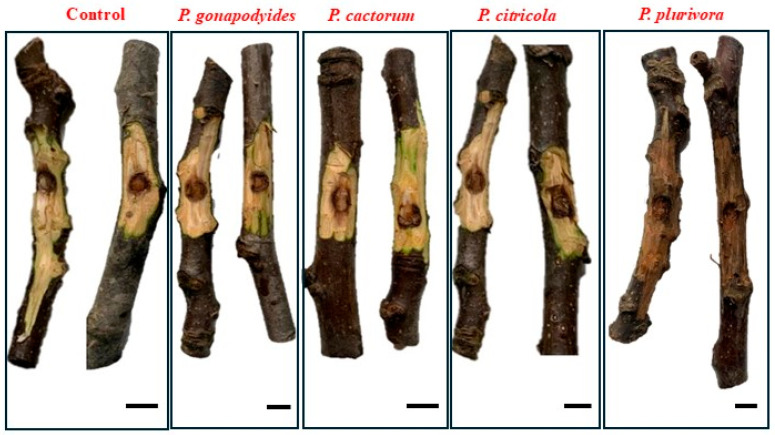
Visual representation of the results of the pathogenicity assays conducted on detached branches of *J. regia*. The image displays ten host segments (from left to right): two negative controls; two branches inoculated with *P. gonapodyides*; two branches inoculated with *P. cactorum*; two branches inoculated with *P. citricola*; and two branches inoculated with *P. plurivora*. It can easily be noticed that the first three species of *Phytophthora* did not induce significant symptoms: necroses remained limited around the infection point, with an extent almost equal to that of the control. In contrast, detached branches inoculated with *P. plurivora* exhibited clearly visible symptoms, with necroses extending along their entire length. All branches were debarked to allow for accurate observation of the lesions and for photographic documentation. Scale bar = 2 cm.

**Table 1 microorganisms-13-02094-t001:** Incidence and relative frequency of *Phytophthora* species isolated from soil and symptomatic *Juglans regia* trees across three surveyed plantations in Tuscany (Italy).

Location (Coordinates)	*P. plurivora*	*P. cactorum*	*P. citricola*	*P. gonapodyides*
**Florence** (43.78624, 11.159425)	100%	0%	0%	0%
**Palaia** (43.583187, 10.801350)	72%	22%	0%	6%
**Cavriglia** (43.540960, 11.470407)	61%	31%	8%	0%

**Table 2 microorganisms-13-02094-t002:** Sequence-characterised representative isolates of *Phytophthora gonapodyides*, *P. cactorum*, *P. citricola*, and *P. plurivora* recovered from root and collar tissues, as well as from rhizosphere soil, in three *Juglans regia* orchards located in Tuscany, central Italy.

Isolate Code	Taxon	Source	Gene Bank Acc. No.
Pg12	*P. gonapodyides*	root tissue, rhizosphere soil	PV987739
Ca1	*P. cactorum*	root tissue, rhizosphere soil	PV987612
Ci24	*P. citricola*	root tissue, rhizosphere soil, collar tissue	PV987611
Plu15	*P. plurivora*	root tissue, rhizosphere soil, collar tissue	PV987766

## Data Availability

The original contributions presented in this study are included in the article. Further inquiries can be directed to the corresponding authors.

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
