# Peer review of "Phytophthora plurivora*: A Serious Challenge for English Walnut (*Juglans regia*) Cultivation in Europe"

_microorganisms, 2025, doi:10.3390/microorganisms13092094_

Round 1

Reviewer 1 Report

Comments and Suggestions for Authors

Comments to the author

The manuscript entitled “Phytophthora plurivora: a serious challenge for English Walnut (Juglans regia) cultivation in Europe” isolated Phytophthora species from three geographically distinct regions and identified Phytophthora plurivora as the primary species causing the most severe disease in English walnut. The authors substantiated their conclusions through molecular analyses and pathogenicity assays. However, some results are not presented in the manuscript, and there are several formatting issues. Specific suggestions are as follows:

-Line 38: Keywords should be arranged in alphabetical order.

-Lines 77–80 and Figure 1 belong to the Results section rather than the Introduction. It is recommended to move these to the Results section or remove them.

-Line 112: Please verify the formula; it should likely be “n/N” rather than “nN.”

-Line 164: MgCl2 should be written as “MgCl₂” (using the correct chemical subscript format).

-Lines 227–274: Please present the results described in text form as figures, for example, by creating a bar chart to illustrate the proportions of Phytophthora species isolated from the three geographically distinct regions.

-Line 316: For Figure 4, please add a scale bar.

In addition, are the walnuts cultivated in the three geographically distinct regions of the same variety? If not, could the proportion of Phytophthora species be influenced by the walnut variety? If different varieties were used, which variety was employed by the authors for the pathogenicity tests?

Author Response

The manuscript entitled “Phytophthora plurivora: a serious challenge for English Walnut (Juglans regia) cultivation in Europe” isolated Phytophthora species from three geographically distinct regions and identified Phytophthora plurivora as the primary species causing the most severe disease in English walnut. The authors substantiated their conclusions through molecular analyses and pathogenicity assays. However, some results are not presented in the manuscript, and there are several formatting issues. Specific suggestions are as follows:

 -Line 38: Keywords should be arranged in alphabetical order.

DONE.

-Lines 77–80 and Figure 1 belong to the Results section rather than the Introduction. It is recommended to move these to the Results section or remove them.

DONE.

-Line 112: Please verify the formula; it should likely be “n/N” rather than “nN.”

DONE.

-Line 164: MgCl2 should be written as “MgCl₂” (using the correct chemical subscript format).

DONE.

-Lines 227–274: Please present the results described in text form as figures, for example, by creating a bar chart to illustrate the proportions of Phytophthora species isolated from the three geographically distinct regions.

DONE. We have accepted your suggestion but felt it would be more informative to report the proportions of Phytophthora species isolated from the three geographically distinct regions in numerical form (percentages). We have therefore created a specific table (Table 1).

-Line 316: For Figure 4, please add a scale bar.

DONE.

In addition, are the walnuts cultivated in the three geographically distinct regions of the same variety? If not, could the proportion of Phytophthora species be influenced by the walnut variety? If different varieties were used, which variety was employed by the authors for the pathogenicity tests?

DONE. They are all populations originating from seed orchards. We have added this information.

Reviewer 2 Report

Comments and Suggestions for Authors

Dear Authors,

       The manuscript entitled "Phytophthora plurivora: a serious challenge for English Walnut (Juglans regia) cultivation in Europe" (Manuscript ID: microorganisms-3829301) has been reviewed. This paper attempted to reveal the pathogen causing dieback on J. regia trees, a severe  disease in Tuscany, central Italy. Four Phytophthora species were isolated from symptomatic J. regia and identified  as P. gonapodyides, P. cactorum, P. citricola and P. plurivora based on the morphological features and molecular analyses. Of these species, P. plurivora was proved to be the main species with overwhelming frequency. 

  1. In Line 20, four Phytophthora species were identified in this work. However, how many isolates were obtained? how many isolates for each Phytophthora species? More  information should be added.
  2. In Line 21, “Among these”should be “Among these species”.
  3. The Abstract seems to be too long. In Line 28-34, these sentences “anthropogenic factors, such as the use of infected plant material and poor management practices...  By virtue of the sustainable business it fuels, many plantations of this species have been established in the region.”might be more suitable to remove to Discussion.
  4. In Line 110, In Line 112, the investigated methodology is very important for this disease.Reference related to the methodology should be cited in this part.
  5. In Line 112,  “DI (%) = (nN)×100 and M (%) = (dN)×100”seems to be error. “nN” should be “n/N”, “dN” should be “d/N”.
  6. In the section of “2.4. Pathogenicity assays”, the pathogenicity assays is not sufficient. The isolates recovered from diseased samples were inoculated to host plants again, thereby completing Koch's postulate.
  7. Figure 1 and Figure 2 both exhibited the symptomscaused by Phytophthora  If the two figure illustrated same topic, please merge them.
  8. In Line 229, “Phytophthora”should be italic.
  9. In Table 1, author provide four isolates.Only four isolates used in this study?
  10. In Figure 3, phylogenetic tree were conducted based on the ITS sequences. To confirm the identification results, it is suggested to provide more genes sequences.
  11. In Line 300-314, Pathogenicity assaydata is not enough to exhibit the pathogenic ability of Phytophthora  Authors should provide more data to reveal the infecting potential of all the tested isolates.
  12. Discussionshould be performed based on the experimental result The irrelevant content should be omitted or removed to Introduction. For example, the detection method such as PCR, LAMP and RPA is not closely irrelevant with this paper.

Author Response

The manuscript entitled "Phytophthora plurivora: a serious challenge for English Walnut (Juglans regia) cultivation in Europe" (Manuscript ID: microorganisms-3829301) has been reviewed. This paper attempted to reveal the pathogen causing dieback on J. regia trees, a severe  disease in Tuscany, central Italy. Four Phytophthora species were isolated from symptomatic J. regia and identified  as P. gonapodyidesP. cactorumP. citricola and P. plurivora based on the morphological features and molecular analyses. Of these species, P. plurivora was proved to be the main species with overwhelming frequency. 

  1. In Line 20, four Phytophthora species were identified in this work. However, how many isolates were obtained? how many isolates for each Phytophthora species? More  information should be added.

Done, we have added this information in Section 2.2. “Phytophthora Isolation and Morphological characterization”.

  1. In Line 21, “Among these” should be “Among these species”.

DONE.

  1. The Abstract seems to be too long. In Line 28-34, these sentences “anthropogenic factors, such as the use of infected plant material and poor management practices...  By virtue of the sustainable business it fuels, many plantations of this species have been established in the region.”might be more suitable to remove to Discussion.

DONE. We have removed the part you indicated, adding only a brief reference to the economic value of walnut cultivation (because this is discussed at length in the Discussion).

  1. In Line 110, In Line 112, the investigated methodology is very important for this disease.Reference related to the methodology should be cited in this part.

DONE.

  1. In Line 112,  “DI (%) = (nN)×100 and M (%) = (dN)×100”seems to be error. “nN” should be “n/N”, “dN” should be “d/N”.

DONE.

  1. In the section of “2.4. Pathogenicity assays”, the pathogenicity assays is not sufficient. The isolates recovered from diseased samples were inoculated to host plants again, thereby completing Koch's postulate.

We did not wrote “Koch's postulates” because we did not use whole plants. However, the use of excised branches to test pathogen virulence is routine in plant pathology, and these protocols have been used with many pathogens, including many species of Phytophthora. See, for example, the following studies: 

Khdiar, M.Y., Burgess, T.I., Scott, P.M., Barber, P.A. and Hardy, G.E.S.J., 2020. Pathogenicity of nineteen Phytophthora species to a range of common urban trees. Australasian Plant Pathology49(6), pp.619-630.

Scanu, B. and Webber, J.F., 2016. Dieback and mortality of Nothofagus in Britain: ecology, pathogenicity and sporulation potential of the causal agent Phytophthora pseudosyringae. Plant Pathology65(1), pp.26-36.

  1. Figure 1 and Figure 2 both exhibited the symptoms caused by Phytophthora  If the two figure illustrated same topic, please merge them.

DONE.

  1. In Line 229, “Phytophthora”should be italic.

DONE.

  1. In Table 1, author provide four isolates. Only four isolates used in this study?

Only four isolates are listed in Table 1 (which has now become Table 2)

 because they corresponded to four distinct morphs, each corresponding to a  Phytophthora species.

  1. In Figure 3, phylogenetic tree were conducted based on the ITS sequences. To confirm the identification results, it is suggested to provide more genes sequences.

Isolate (Phytophthora species) identification, based on ITS sequence analysis, was fully complemented by the analyses of: 1) macro-morphological; 2) micro-morphological; and 3) physiological (growth/temperature relationships) characters. In addition, the isolates (species) reported in this study were compared with reference isolates present in public mycological collections (ex-types) and with isolates (species) already present in our own collection.

  1. In Line 300-314, Pathogenicity assay data is not enough to exhibit the pathogenic ability of Phytophthora.  Authors should provide more data to reveal the infecting potential of all the tested isolates.

We answered this question in point 6 above. The infecting potential is clearly demonstrated by the tests conducted. On the other hand, these species are well known in the literature for their aggressiveness. Our study confirms what has been reported by other authors on the high aggressiveness of Phytophthora plurivora towards various hosts and on the reduced or moderate virulence of the other species.

  1. Discussion should be performed based on the experimental result. The irrelevant content should be omitted or removed to Introduction. For example, the detection method such as PCR, LAMP and RPA is not closely irrelevant with this paper.

This is a short sentence of just a few lines, which could easily be eliminated. However, precisely for the reasons you highlighted in the previous point 10 (where you raised the issue of the appropriateness of taxon identification), we believe it is important to include - among the control strategies against this disease - the use of molecular methods for rapid identification. As we wrote in the following sentence, “The integration of molecular diagnostics into regular monitoring programmes would facilitate timely intervention and containment strategies”.

Reviewer 3 Report

Comments and Suggestions for Authors

To the Editor,

The manuscript titled "Phytophthora plurivora: a serious challenge for English Walnut (Juglans regia) cultivation in Europe" by Alessandra Benigno et al. provides a significant contribution to the understanding of Phytophthora plurivora as an emergent pathogen in walnut cultivation and is relevant for both plant pathology and agricultural sustainability.

The manuscript is clear, novel, and an important study for potential readers, because it combines morphological, molecular, and pathogenicity assays to confirm that Phytophthora plurivora is the primary pathogenic agent of walnut decline in Tuscany. This pathogen is becoming more and more a pressing agricultural and ecological threat in Europe and the Mediterranean basin. The attached manuscript contains some improvement suggestions.

To improve the manuscript, I recommend that the authors expand the discussion of climate projections and management strategies (e.g., biocontrol and resistant cultivars—more recent research) and improve figure legends to be more descriptive and to ensure that all species, genera, and family names are italicised (including the reference list).

Author Response

To the Editor,

The manuscript titled "Phytophthora plurivora: a serious challenge for English Walnut (Juglans regia) cultivation in Europe" by Alessandra Benigno et al. provides a significant contribution to the understanding of Phytophthora plurivora as an emergent pathogen in walnut cultivation and is relevant for both plant pathology and agricultural sustainability.

The manuscript is clear, novel, and an important study for potential readers, because it combines morphological, molecular, and pathogenicity assays to confirm that Phytophthora plurivora is the primary pathogenic agent of walnut decline in Tuscany. This pathogen is becoming more and more a pressing agricultural and ecological threat in Europe and the Mediterranean basin. The attached manuscript contains some improvement suggestions.

Thank you for your appreciation of our manuscript. We have accepted almost all of your suggestions. Regarding your suggestion to write family names in italics, we would like to point out that, according to taxonomic convention, family names should not be written in italics (unlike genera, species, etc.).

To improve the manuscript, I recommend that the authors expand the discussion of climate projections and management strategies (e.g., biocontrol and resistant cultivars—more recent research) and improve figure legends to be more descriptive and to ensure that all species, genera, and family names are italicised (including the reference list).

We have expanded the section on the impact of climate change and how to deal with the problem slightly. We did not consider extending the Discussion further, partly because the other referees objected that it was too long. Regarding your suggestion to discuss biocontrol and resistant cultivars, please note that we had already mentioned the issue of selecting genetically resistant plant material and biocontrol as a possible control strategy (see citation no. 54).

Reviewer 4 Report

Comments and Suggestions for Authors

The manuscript provides valuable information on the identification of a pathogen responsible for walnut dieback. The text is generally well written. I have only few minor comments and some optional comments. Optional comments do not have to be implemented by the Authors.

Minor comments:

Introduction, lines 48-51: this sentence requires linguistic correction.

Line 112: Please check if the formulas are correct. There might be something missing. Shouldn’t the number of symptomatic or dead trees be divided by the total number of trees?

Line 114: add “number of” to explanation of each symbol.

Section 2.2: In the end of this section it is stated that isolates were identified as P. gonapodyides, P. cactorum, P. citricola and P. plurivora. This statement requires a citation of taxonomic literature/taxonomic keys describing morphology of these species.

Line 163: provide manufacturer of Taq DNA Polymerase.

Section 3.2: It would be good to specify the number of isolates obtained in each location.

Line 231: replace “isolated” with “isolates”

Line 461: finish the sentence, write were sequence data was deposited.

Table below line 465 should be deleted. It is not necessary to provide links to the NCBI records. Accession numbers are sufficient and they are provided in Table 1.

Optional comments:

Abstract summarizes research outcomes in a very general way. I believe it would be beneficial for future readers if more details on the methods and results appeared in this part of the paper. Why not write in more detail about the good work that was actually done? It may encourage more people to read this paper. It could be specified for example: which morphological structures were measured, which region was sequenced, how many Phytophthora isolates were obtained and what percentage of these isolates was identified as P. plurivora, … etc. Discussion points in the last 9 lines of the abstract can be shortened.

Section 3.2: It would be good to present some photos of the structures/colonies of the four Phytophthora species. Consider adding a figure if you have such photos.

I also wonder about the differences in growth rate among the four Phytophthora spp. You measured colony diameters at different temperatures and reported which temperature was optimal for mycelial growth for each species. The four species have more or less the same optimum. However, I wonder if there was a statistically significant difference in colony diameter or growth rate among the species? If yes, consider presenting data in additional table. P. plurivora is supposed to be more competitive (as you write in the discussion), so maybe it grows faster?

Author Response

The manuscript provides valuable information on the identification of a pathogen responsible for walnut dieback. The text is generally well written. I have only few minor comments and some optional comments. Optional comments do not have to be implemented by the Authors.

Minor comments:

Introduction, lines 48-51: this sentence requires linguistic correction.

Done. We have modified and simplified the sentence.

Line 112: Please check if the formulas are correct. There might be something missing. Shouldn’t the number of symptomatic or dead trees be divided by the total number of trees?

Done.

Line 114: add “number of” to explanation of each symbol.

Done.

Section 2.2: In the end of this section it is stated that isolates were identified as P. gonapodyides, P. cactorum, P. citricola and P. plurivora. This statement requires a citation of taxonomic literature/taxonomic keys describing morphology of these species.

Done.

Line 163: provide manufacturer of Taq DNA Polymerase.

Done.

Section 3.2: It would be good to specify the number of isolates obtained in each location.

Done.

Line 231: replace “isolated” with “isolates”

Done.

Line 461: finish the sentence, write were sequence data was deposited.

Done.

Table below line 465 should be deleted. It is not necessary to provide links to the NCBI records. Accession numbers are sufficient and they are provided in Table 1.

We prefer to leave it because in another our article published in Microorganisms (Benigno, A., Aglietti, C., Cacciola, S.O. and Moricca, S., 2025. Microorganisms, 13(3), p.567), we were specifically asked to provide links to the NCBI records.

 Optional comments:

Abstract summarizes research outcomes in a very general way. I believe it would be beneficial for future readers if more details on the methods and results appeared in this part of the paper. Why not write in more detail about the good work that was actually done? It may encourage more people to read this paper. It could be specified for example: which morphological structures were measured, which region was sequenced, how many Phytophthora isolates were obtained and what percentage of these isolates was identified as P. plurivora, … etc.

Done. Thank you for this meaningful suggestion.

Discussion points in the last 9 lines of the abstract can be shortened.

Done.

Section 3.2: It would be good to present some photos of the structures/colonies of the four Phytophthora species. Consider adding a figure if you have such photos.

I also wonder about the differences in growth rate among the four Phytophthora spp. You measured colony diameters at different temperatures and reported which temperature was optimal for mycelial growth for each species. The four species have more or less the same optimum. However, I wonder if there was a statistically significant difference in colony diameter or growth rate among the species? If yes, consider presenting data in additional table. P. plurivora is supposed to be more competitive (as you write in the discussion), so maybe it grows faster?

We have not included additional data (photos of structures/colonies, growth rates, etc.) because this is not a taxonomic article but rather a study focused on the pathogenicity and role of P. plurivora in the development of the disease on walnut trees. On the other hand, this information has already been extensively described in previous studies, which were cited in our article.

Round 2

Reviewer 1 Report

Comments and Suggestions for Authors

The author modified all the content I mentioned.